# Physical Exercise as Disease-Modifying Alternative against Alzheimer’s Disease: A Gut–Muscle–Brain Partnership

**DOI:** 10.3390/ijms241914686

**Published:** 2023-09-28

**Authors:** Debora Cutuli, Davide Decandia, Giacomo Giacovazzo, Roberto Coccurello

**Affiliations:** 1Department of Psychology, University of Rome La Sapienza, 00185 Rome, Italy; davide.decandia@uniroma1.it; 2European Center for Brain Research, Santa Lucia Foundation IRCCS, 00143 Rome, Italy; giacomo.giacovazzo@gmail.com; 3Facoltà di Medicina Veterinaria, Università degli Studi di Teramo (UniTE), 64100 Teramo, Italy; 4Institute for Complex Systems (ISC), National Council of Research (CNR), 00185 Rome, Italy

**Keywords:** physical exercise, Alzheimer’s disease, neuropsychiatric symptoms, microbiota, gut–muscle–brain axis, irisin, BDNF

## Abstract

Alzheimer’s disease (AD) is a common cause of dementia characterized by neurodegenerative dysregulations, cognitive impairments, and neuropsychiatric symptoms. Physical exercise (PE) has emerged as a powerful tool for reducing chronic inflammation, improving overall health, and preventing cognitive decline. The connection between the immune system, gut microbiota (GM), and neuroinflammation highlights the role of the gut–brain axis in maintaining brain health and preventing neurodegenerative diseases. Neglected so far, PE has beneficial effects on microbial composition and diversity, thus providing the potential to alleviate neurological symptoms. There is bidirectional communication between the gut and muscle, with GM diversity modulation and short-chain fatty acid (SCFA) production affecting muscle metabolism and preservation, and muscle activity/exercise in turn inducing significant changes in GM composition, functionality, diversity, and SCFA production. This gut–muscle and muscle–gut interplay can then modulate cognition. For instance, irisin, an exercise-induced myokine, promotes neuroplasticity and cognitive function through BDNF signaling. Irisin and muscle-generated BDNF may mediate the positive effects of physical activity against some aspects of AD pathophysiology through the interaction of exercise with the gut microbial ecosystem, neural plasticity, anti-inflammatory signaling pathways, and neurogenesis. Understanding gut–muscle–brain interconnections hold promise for developing strategies to promote brain health, fight age-associated cognitive decline, and improve muscle health and longevity.

## 1. Introduction

An expanding body of evidence has repeatedly confirmed the view that physical activity (PA) or physical exercise (PE) can be an effective, non-pharmacological, therapy-like approach to improve the management of clinical symptoms in the category of mental illness, especially for depressive and anxiety disorders [1,2,3]. However, a first and preliminary distinction should be made between PA and PE. Despite both terms being used interchangeably, for PA we mean any motor activity or body movement requiring muscle contraction and thus energy utilization, whereas PE is a planned, repetitive, schematized/structured, and purposeful PA [4]. The positive outcome of PE intervention is now recognized to be beneficial in different neurological/neurodegenerative diseases, reducing the risk of premature mortality in depression [5], as well as dementia and mild cognitive impairment (MCI) [6], and attenuating cognitive decline progression in severe forms of dementia such as Alzheimer’s disease (AD) [7,8]. Interestingly, the adequacy and efficacy of PE both in the area of psychiatric diseases (i.e., schizophrenia and major depressive disorder) [5,9] and impaired cognition (dementia and AD) [10] disclose the possibility that a regular PE might also favorably impact neuropsychiatric symptoms (NPSs) in AD. Long undervalued, the clinical relevance of NPSs in AD is now widely acknowledged [11]. Indeed, NPSs are not only invalidating symptoms affecting the quality of daily living in MCI and AD patients, but are also a proving ground for the evaluation of clinical severity and disease prognosis, which is generally worse in the case of co-occurrence of cognitive deterioration and NPSs [12]. Interestingly, it has been observed that some NPSs such as depression, apathy, impulsivity, and irritability can manifest before cognitive decline, and for this reason can be clinically identified to predict rapid progression to MCI and medial temporal lobe atrophy, and faster MCI conversion to AD [13,14,15].

## 2. Alzheimer’s Disease and Neuropsychiatric Symptoms

As is widely known, AD is the most common cause of dementia and its prevalence is expected to increase worldwide from 50 million people in 2010 to 113 million people in 2050 [16]. AD is a slow progressive neurodegenerative disease which encompasses molecular, cellular, neural circuitry, and cognitive dysregulations. The pathophysiology is characterized mainly by the accumulation of Tau protein in the brain, especially in the medial temporal lobe and neocortical structures forming intracellular neurofibrillary tangles, and the aggregation of amyloid-β (Aβ) peptide forming extracellular amyloid plaques, together with the overactivation of glia and the loss of synaptic homeostasis, neurons, or neuronal network integrity [17]. To date, there is still no cure for AD and pharmacological approaches focus more on reducing symptoms using cholinesterase inhibitors such as donepezil, rivastigmine, and galantamine, as well as N-methyl d-aspartate (NMDA) antagonists such as memantine [18]. According to the leading amyloid cascade hypothesis, the progressive and toxic accumulation of extracellular Aβ peptides, formation of β-amyloid fibrils, and defective Aβ clearance are the major causes of AD pathogenesis [19]. However, especially because the recurring disappointments concerning anti-Aβ therapy and the failure of a large number of clinical trials addressing Aβ aggregation, there is a need to consider the recent strategy focused on passive (i.e., antibodies) and active (i.e., vaccines) immunization, and the hope raised by the anti-amyloid and anti-tau immunotherapy in AD [20]. Although many problems of immunotherapy such as partial efficacy, adverse effects (i.e., safety), and target selectivity have to be resolved [21], there are few doubts that this approach holds new promises for the development of disease-modifying interventions. Thus, several antibodies have been used both in preclinical studies using transgenic AD-like mice [22,23,24], in mouse models of sporadic AD [25], as well as in immunotherapy trials [26]. Immunotherapies with recent approval from the Food and Drug Administration are aducanumab and lecanemab, which, in spite of safety concerns and controversial clinical outcomes [26,27], are third-generation, high-clearance, anti-amyloid immunotherapies. These are considered a step forward in AD therapy and a significant improvement in the modest risk/benefit ratio has been shown by the pharmacological approaches available so far.

AD is characterized by prominent amnestic cognitive impairment (with a lower prevalence of patients with non-amnestic cognitive impairment) and difficulties in expressive speech, visuospatial processing, and executive functions [17]. In AD patients, the severity of cognitive decline is heterogenous, and it could start with cognitive difficulties in the absence of impaired performance following cognitive testing. MCI refers to the earliest manifestation of cognitive decline in which a single domain, or multiple cognitive domains, presents an impairment, but the overall functionality is preserved [28].

In AD patients, cognitive decline is mostly accompanied by NPSs, which are referred to as behavioral and psychological symptoms of dementia (BPSD) and include a wide range of manifestations such as apathy, depression, anxiety, psychosis, agitation, aggression, sleep disturbances, and other problematic behaviors such as wandering, sexually inappropriate behaviors, and care refusal [29,30]. These symptoms have a deleterious impact on the patient’s quality of life. In fact, they cause emotional distress, functional impairment, frequent hospitalizations, and earlier mortality [31]. The prevalence of NPSs in AD patients is between 56 and 98% in the community and could rise up to 91–96% in hospitals or long-term care facilities [32]. Given the heterogeneity of these symptoms, their high prevalence, and their detrimental impact on both psychological well-being and overall quality of life, patients suffering from AD-associated NPSs are often prescribed broad-spectrum pharmacological therapies. Unfortunately, these treatments often prove to be inadequate at improving the patients’ conditions, as they frequently entail various adverse side effects. For instance, in the case of atypical antipsychotics, several studies have shown their liability to induce severe metabolic alterations and even elevate the risk of mortality among patients with dementia [33,34].

Apathy, aggression, and depression are correlated with age, disease progression, and cognitive impairment in AD patients. Aggressive behaviors are more common in males than females [35]. NPSs exacerbate cognitive decline in AD patients, and this is a burden and a source of stress not only for patients but also for caregivers [36]. Heavy caregiver burdens not only led to financial difficulties but also social isolation, health deterioration, and psychological disorders such as depression due to the problems associated with the management of the clinical situation.

Apathy is the most common NPS associated with AD and refers to a lack of motivation in almost one out of three domains, including goal-directed behavior, goal-directed cognitive activity, and emotion. In comparison to psychosis or anxiety, apathy has a high level of persistence [37]. The apathy symptomatology was found to be correlated with morpho-functional alterations in the anterior cingulate circuit, involved in motivation, and the medial orbitofrontal circuit which is involved in the integration of visceral–amygdalar functions [29]. Depression is the second most common NPS in AD patients and it is associated with an increased risk of developing AD pathology, and a higher risk of conversion from MCI to AD [38], suggesting that depression could be a prodromal feature of AD [39]. Neuroimaging studies correlated AD and late-life depression with a bilateral reduction in the hippocampal volume [40]. Increased cortisol production can link depression and dementia by inducing hippocampal atrophy [41]. In fact, depressive symptoms activate the Hypothalamic–Pituitary–Adrenal (HPA) axis and increase glucocorticoid levels, which may in turn damage the hippocampus and result in a down-regulation of hippocampal glucocorticoid receptors, causing the impairment of negative feedback to the HPA axis and the glucocorticoid cascade [42]. These effects may lead to hippocampal atrophy and cognitive decline [41].

Agitation and aggression include excessive motor activity (pacing, restlessness), verbal aggression (yelling, screaming), and/or physical aggression (scratching, grabbing, slamming doors). These behaviors are associated with emotional distress such as changes in mood or irritability and lead to a significant impairment in social functioning, daily activities, and interpersonal relationships [43]. Many studies have highlighted the correlation between agitation and loss of brain volume in brain regions such as the frontal cortex, anterior cingulate cortex, posterior cingulate cortex, insula, amygdala, and hippocampus [44,45,46], and decreased cholinergic activity in the frontal and temporal cortex and decreased serotonin [47]. Delusions and hallucinations are frequently reported and these psychotic symptoms could significantly affect the psychological well-being of patients, causing the aggressive behavior that could in turn increase the risk of institutionalization. Moreover, hallucinations and delusions are strongly associated with rapid cognitive decline. Psychotic symptoms are related to atrophy in the lateral frontal, lateral parietal, and anterior cingulate gyrus, implicated in the cortico-subcortical networks and in the regulation of complex human behaviors [48]. Furthermore, neuroimaging studies showed that psychotic symptoms and agitation can overlap with one another [45] (Figure 1).

Of particular interest, it has been observed that some NPSs such as depression, apathy, impulsivity, and irritability can manifest before the beginning of cognitive decline, and, for this reason, can be clinically identified to predict rapid progression to MCI and medial temporal lobe atrophy, as well as faster conversion from MCI to AD [14,15]. In fact, MCI patients with NPS comorbidities displayed an annual conversion rate of the pathology to AD of ∼21% [49]. Many preclinical studies have shown that NPSs could even be found in widely used animal models of AD. In a detailed review [50], researchers described the presence of behavioral alterations in transgenic mouse models of AD corresponding to some BPSD found in AD patients. Social withdrawal and depressive-like behaviors were found to be related to the progression of AD pathology in mouse models of AD. Interestingly, at early stages, some models displayed both aggression and sleep–wake alterations [50]. All these behavioral alterations reported in different AD mouse models confirm the existence of features corresponding to AD pathophysiology.

## 3. Physical Exercise, Gut Microbiota, and Alzheimer’s Disease Risk

### 3.1. The Role of Physical Exercise for Brain Health and Alzheimer’s Disease Prevention

There is a robust consensus about the fact that a non-excessive level of PE and long-term exercise intervention can positively contribute to reducing chronic systemic inflammation, ameliorating human health, and promoting longevity [51,52,53]. Remarkably, accumulating evidence supports the view that the benefits of PE may also extend to patients with neurodegenerative disorders, particularly those with AD [54]. Indeed, in addition to other factors (such as aging, low education, and poor diet), low levels of PE or complete physical inactivity are considered crucial risk factors for developing dementia and AD [55]. Converging reports addressing the benefits of PE for reducing AD risk, and the favorable impact of PE on brain aging and preservation of cognitive function [56] have underlined the key importance of different types of long-term exercise training for the prevention of the AD trajectory [56,57]. Thus, regular exercise is much more successful as a neuroprotective strategy than sporadic or occasional activity, including aerobic and resistance/strength exercise [56,57,58]. Several studies have investigated the multiple mechanisms that might account for the beneficial impact of PE either on the likelihood to develop dementia and AD or as disease-modifying option to reduce cognitive deterioration and delay the onset of memory loss. Within this frame, the study of hippocampal physiology has received particular attention, with studies reporting an increased size of the hippocampus following PE such as aerobic exercise in both adults [59] and older adults [60]. In addition to the increase in brain volume, found to be particularly enlarged in the frontal lobes of participants undergoing a 6-month aerobic exercise program [61], the main changes triggered by regular training programs involve non-mutually exclusive mechanisms such as an increase in cerebral blood flow and its positive effects on cognition (i.e., executive function) [62], neurogenesis [63], angiogenesis [64,65], neurotrophic factors [66], and neuroplasticity [67,68]. In a recent systematic review of the literature focused on the effects of PE on the mechanisms underlying AD etiology [69], the authors identified about eight principal pathways through which PE can modify AD pathophysiology. Among these pathways, it is worth remembering the roles of the immune system and inflammation that are, in turn, associated with mechanisms of cell survival or cell senescence, as well as with the protective role of PE against oxidative stress, lipid peroxidation, and improvements in energy metabolism and insulin sensitivity [70,71] (Figure 2).

### 3.2. Exploring the Impact of Gut Microbiota Dysbiosis and Neuroinflammation in Alzheimer’s Disease Pathogenesis

In recent years, a great deal of interest has been focused on the deleterious impact produced by GM dysbiosis, the growth of pro-inflammatory bacteria, increased permeability of the intestinal barrier, and systemic inflammation in AD pathogenesis [72,73], including the comorbidity of NPSs and AD [74].

Various studies have led to the formulation of a neuroinflammation theory of AD, proposing a central role of the immune system, specifically of astrocytes and microglia, the overactivation of which, caused by the presence of insoluble Aβ oligomers, would up-regulate the production of pro-inflammatory agents causing neuronal damage and cell death [75,76,77,78]. In the first formulation of this idea, indications were made exclusively to the inflammation of the central nervous system (CNS), but more recent studies have suggested the involvement of the peripheral nervous system (PNS) as a concomitant factor for the increase in neuroinflammation, as is typically observed in neurodegenerative disorders [79,80]. In the human body, the intestinal tract has the largest microecosystem, defined as GM, which counts over 100 trillion microorganisms (10^14^) including more than 2000 known different species of bacteria [81]. A healthy GM is characterized by bacterial stability and species diversity. In the GM, bacteria are classified by genus, family, order, and phylum. *Firmicutes* (such as *Lactobacillus*) and the *Bacteroidetes* represent the main bacterial phyla in the gut, accounting for 90% to 95% of the total microbiota, followed by *Proteobacteria*, *Actinobacteria* (such as *Bifidobacterium*), and *Cyanobacteria* [82,83]. The phylum *Firmicutes* is made up of more than 200 genera consisting of *Clostridium*, *Blautia*, *Faecalibacterium*, *Enterococcus*, *Lactobacillus Eubacterium*, *Roseburium*, and *Ruminococcus*. *Bacteroidetes* consist of different genera like *Bacteroides* and *Prevotella* [84]. The most common bacteria in the stomach are *Lactobacilli*, *Veillonella*, and *Helicobacter*, while in the duodenum, jejunum, and ileum there is a much higher concentration of *Bacilli*, *Streptococcaceae*, *Actinomycinaeae*, and *Corynebacteriaceae*. *Lachnospiraceae* and *Bacteroidetes* are the more prominent bacteria found in the colon [85].

The bidirectional crosstalk between the brain and gut, known as the “gut–brain axis”, is mainly based on the direct and indirect connection of GM with the CNS, the autonomic nervous system (ANS), the enteric nervous system (ENS), and the HPA axis [86], and it involves multiple overlapping pathways, including the neuroendocrine and immune systems, playing a key role in neuronal development, brain function, cognitive regulation, and aging [87]. The gut–brain axis influences fundamental brain processes, such as neuroinflammation, activation of the HPA axis, neurotransmission, and neurogenesis, even modulating complex behaviors [88]. In the gut, microbes metabolize complex carbohydrates producing short-chain fatty acids (SCFAs) such as acetate, propionate, and butyrate (more than 95% of the total SCFAs), which can be utilized as energy. They could act as signaling molecules involved in lipid metabolism and glucose/insulin regulation, important for the maturation of the microglia [89]. SCFAs could indirectly affect even neurotransmission by modulating the synthesis of different neurotransmitters with a direct impact on cognition and behavior. In particular, butyric and propionic acids enhance the synthesis of dopamine, noradrenaline, and serotonin, eliciting the expression of tyrosine and tryptophan hydroxylase [90]. Over the years, there has been an increase in scientific reports about the involvement of GM in the regulation of several physiological functions that have a strong effect on the general state of health of individuals. Thus, it becomes increasingly clear how important it is to preserve the delicate balance of GM. Microbial differences can depend on age, sex, body mass index, host genotype, but the microecological balance is strongly affected by different lifestyle factors such as dietary patterns, smoking, alcohol, antibiotics, drugs, toxins, and pathogens [91]. A GM imbalance, namely a dysbiosis, was found in several neurological disorders including AD, Parkinson’s disease, and major depressive disorder [88,92]. GM dysbiosis may lead to systemic inflammation, which determines microglia overactivation and blood brain barrier (BBB) damage, causing the deleterious crossing of pathogens and immune cells [93]. Disruptions to gut barrier integrity may spur an influx of lipopolysaccharide (LPS) into the host systemic circulation [94]. LPS is a highly acylated saccharolipid and structural component of the outer membrane of Gram-negative bacteria. It is a potent stimulant of the host immune response and secretion of pro-inflammatory cytokines and chemokines because it is sensed by the body’s innate immunity, thus alerting it about potential threats of invasion by pathogens [95]. LPS is detected by toll-like receptors (TLRs) such as the TLR4 which is a transmembrane receptor that belongs to the pattern recognition receptor (PRR) family and is expressed in different cells modulating the innate immune response (microglia, astrocytes, macrophages, and leukocytes) [96]. Once LPS is detected, the TLR4 activates the nuclear factor kappa B (NF-κB) pathway, eliciting the production of pro-inflammatory mediators such as inducible nitric oxide synthase (iNOS), Cyclooxygenase-2 (COX-2), Interleukin-1β (IL-1β), and Tumor Necrosis Factor alpha (TNF-α) by the immune system cells [97].

The latest research is focusing on the GM dysbiosis found in AD patients in order to explore how the gut–brain axis is involved in a condition of up-regulated neuroinflammation. In a study by Cattaneo and co-workers, an increase in the *Escherichia*/*Shigella* bacterial genera was found, which are known for mediating inflammation, in the fecal samples of AD patients, together with an increase in the expression of proinflammatory cytokines IL-1β and CXCL2 in the blood [98]. Different studies reported an increase in *Escherichia*/*Shigella*, *Bacteroides*, and *Ruminococcus* and a decrease in *Eubacterium rectale*, *Bifidobacterium*, and *Dialister* in AD patients and aged individuals with cognitive impairment [99,100,101]. Animal models of AD confirmed the GM dysregulation associated with AD pathology. In fact, in a transgenic model of AD, the 5xFAD mice, characterized by the rapid development of amyloid plaques in the brain, increased levels of Aβ were found also in the gastrointestinal (GI) system, together with an increase in the *Firmicutes*/*Bacteroidetes* ratio [102]. To further prove the connection between the gastrointestinal system and Aβ burden, it was found that fecal transplantation from wild-type mice to AD-like mice models was able to improve cognitive functions, reducing the formation of amyloid plaques and neurofibrillary tangles and diminishing glial reactivity [103]. Furthermore, the peripheral immune activation was found to be down-regulated with a decreased intestinal macrophage activity and lower presence of circulating blood inflammatory monocytes [103]. In a rat model of AD, obtained by injecting the Aβ_1-42_ peptide in the hippocampus, an important GM alteration with an increase in pro-inflammatory bacteria, a reduction in anti-inflammatory bacteria, and the consequent stimulation of the immune system were found [104]. Moreover, in the same study, the administration of fructooligosaccharides was able to increase the probiotic *Lactobacillus*, the anti-inflammatory *Bifidobacterium*, and other bacteria which stimulated the production of acetylcholine, dopamine, serotonin, and norepinephrine in the brain, thus alleviating cognitive decline in AD-like animals [104]. In fact, some bacterial genera such as *Escherichia coli*, *Bacteroides*, *Eubacterium*, and *Bifidobacterium* are involved in the production of neurotransmitters such as acetylcholine, GABA, and glutamate, and alterations in these were found in AD patients [105,106]. Moreover, metabolomic analysis on amnestic MCI and AD patients showed the dysregulation of tryptophan metabolism with a reduction in 5-HTP [74,107]. Interestingly, by using the pharmacological AD-like mouse model of Aβ_1-42_ peptide delivery into the lateral ventricle, both an alteration in GM and the inhibition of cholinergic anti-inflammatory patterns by the reduction in choline acetyltransferase (CHAT) expression in the colon, and a parallel decrease in the expression of M1 acetylcholine receptor in hippocampus and forebrain [108], have been shown. This study underlines the importance of the vagus nerve-mediated cholinergic signaling in the mediation of gut–brain functional communication and its relevance for the exacerbation of AD pathophysiology. Notably, GM alterations in AD mouse models appear to be sex-specific. In fact, a study showed increased *Prevotella* and *Ruminococcus* but reduced *Sutterella* abundance in female mice [109]. In another study, in App^NL-G-F^ female mice, an increased abundance of *Bacteroides*, *Alistipes*, *Turicibacter*, *Ruminococcus*, *Romboutsia* and *Akkermansia* was found, which positively correlated with increased astrogliosis [110]. The GM diverges according to gender from early life, showing differences in composition and alpha-diversity. Notably, boys and girls exhibit significant variances in *Actinobacteria*, *Firmicutes*, and *Bacteroidetes* phyla, with boys having a higher *Bacteroidetes*/*Firmicutes* ratio. Although it may seem simplistic, the current hypothesis is that dynamic hormonal changes, such as the early postnatal testosterone rise in males or the onset of puberty in both sexes, may contribute to these disparities, but ongoing investigations in this field are currently being conducted [111].

### 3.3. A Complex Connection: Neurotransmitter Deregulation, Gut–Brain Axis, and Neurological Disorders

As previously mentioned, the deregulation of neurotransmitters, such as glutamate, acetylcholine, dopamine, GABA, serotonin, and norepinephrine, is extensively described in AD [112,113,114,115], strengthening the concept of the gut–brain axis and its implication in neurological disorders. Alterations in these neurotransmitters can be found in patients suffering from anxiety [116,117] or depression [118], which are some of the NPSs found in AD patients [12,14]. Anxiety can alter gastrointestinal function, and anxiety is a mental condition that can be elicited by GM dysbiosis, as demonstrated by a correlation between intestinal infection and the development of an anxiety disorder [119]. Rodent studies confirmed the connection between GM alterations and anxiety symptoms. In fact, germ-free (GF) mice exhibit reduced anxiety-like behaviors together with an increase in the hippocampal levels of serotonin and a much higher plasma concentration of tryptophan. Of particular interest was the normalization of anxiety levels following the restoration of the gut microbial population in these mice [120]. A systematic review by Yang et al. evaluated a total of 21 studies present in the literature, the aim of which was to reduce anxious symptoms following the regulation of the intestinal microbiota. In 15 studies, probiotics were used, while the remaining 6 studies focused on the use of other strategies, such as, for example, the remodulation of dietary patterns. The authors found that more than half of the studies reported beneficial effects on anxious symptoms and most of the beneficial effects were provided by strategies that did not include the use of probiotic supplementation [121].

There is increasing evidence that depression could be linked to GM alterations, strengthening the involvement of the gut–brain axis in this mood disorder [74,122,123,124]. Cross-sectional studies reported an increasing abundance of *Actinobacteria* and decreased levels of *Bacteroidetes*, and mixed results on *Firmicutes* and *Proteobacteria* in subjects with major depressive disorder [125,126]. Probiotics with or without antidepressants showed some results in alleviating depressive symptoms through the attempt to restore a healthy GM [127]. Although the authors suggest a possible anti-inflammatory mechanism exerted by probiotics in alleviating depressive symptoms, there are still some unresolved problems, such as the implication of BBB damage or intestinal hyperpermeability for the evaluation of this effect. For example, human and animal studies found that depression was associated with an increase in intestinal permeability and bacterial translocation, leading to immune response, thus supporting the “leaky gut” hypothesis [128,129].

In depressed patients, several studies reported an increase in proinflammatory cytokines, such as IL-1, IL-6, IL-8, IL-12, TNF-α, and decreased levels of anti-inflammatory cytokines, such as IL-10, together with a hyperproduction of reactive oxygen species (ROS) and reactive nitrogen species (RNS), with resulting damage to cell membranes, proteins, mitochondria, and DNA [128,130]. Nevertheless, the mechanisms underlying the role of the gut–brain axis in depression and in other neurological disorders are not fully understood, and the scientific community, in recent years, has increased its efforts to find more accurate theories of this relationship.

### 3.4. Impact of Physical Exercise on the Gut–Brain Axis and Neurological Disorders: An Overlooked Role

Despite the level of knowledge gathered about the changes occurring in the gut microbial population and risk of AD or neurodevelopmental disorders [72,73,74,124], as well as about the tight relationship between PE and the colonization of health-promoting bacterial taxa [131], there is only sporadic information concerning the beneficial effects exerted by PE on the gut–brain axis and AD progression [131,132]. Since many detailed descriptions either of the gut microbial ecosystem or of the entire genome of gut microorganisms (i.e., the microbiome) have already been provided [74,124,133], no further explanation of GM and its communication pathways, main composition, and characteristics will be considered in this context. Thus, according to the present scenario, the functional contribution of skeletal muscle (SM) to the equation linking the gut–brain axis, PE, dementia, and NPSs in AD has been frequently undervalued or neglected. Several pieces of evidence support the view that non-extreme or strenuous exercise promotes beneficial changes in microbial composition via, for instance, an increase in the concentration of selected SCFAs such as butyrate (C4) [134], and that an increase in butyrate-producing bacteria protecting the gastrointestinal epithelium (e.g., *Faecalibacterium prausnitzii*) is quantifiable in trained animals [135]. Consistently, the GM of exercised individuals has been shown to display increased microbial diversity and butyrate-producing taxa (e.g., *Coprococcus*, *Roseburia Clostridiales*, *Lachnospiraceae*, and *Erysipelotrichaceae*) [136]. Interestingly, well-trained subjects such as athletes show an increased abundance of *Akkermansia muciniphila* (*A*. *muciniphila* from *Verrucomicrobia* phylum) in their GM, which is linked to the fact that colonization by *A*. *muciniphila* is conversely reduced in both depression and AD [74]. Notably, *A*. *muciniphila* is a mucin-degrading bacterium whose colonization of the mucosal layer is associated with the prevention of an age-dependent decrease in mucus layer thickness and intestinal inflammation [137]. Moreover, it is known that obese individuals display a high *Firmicutes* to *Bacteroides* ratio (two most abundant phyla in GM) [138], while PE has been shown to reduce the gut abundance of the butyrate-producer *Firmicutes* phylum in a model of diet-induced obesity [139,140], thus depicting a “gut-mediated” mechanism by which PE can contribute to obesity prevention. Interestingly, the dichotomy between voluntary and forced PE is emblematic of the functional impact that PE may have on GM ecology and also SM physiology. Indeed, in the GM of animals that underwent forced exercise (i.e., treadmill running), it was found an overgrowth of bacteria such as *Ruminococcus gnavus* [141], which is involved in intestinal mucosa degradation and because of that is considered responsible for intestinal inflammation (e.g., intestinal bowel syndrome and Crohn’s disease) [142].

## 4. Gut Microbiota-to-Skeletal Muscle Axis

A first point of conjunction between the modulation of GM diversity and SM function and metabolism can probably be identified in lactate production and in its conversion into propionate (C3), which, together with butyrate (C4) and acetate (C2), is one of the major SCFA microbial metabolites in the GI tract [134]. The “lactate-propionate” shuttle can be considered a bidirectional process involving the gut–muscle axis. Indeed, lactate in the gut can be produced by probiotic microorganisms such as *Lactobacillus* and *Bifidobacterium* and then converted to SCFAs [143], while, at the same time, it can be also produced through muscle contraction and transported into the gut from the bloodstream to be utilized by lactate-utilizing bacteria, providing the energy substrate for PE [144].

Interestingly, excessive lactate content in the gut may have detrimental effects due to its effect on muscle deterioration and the derangement of heart function through microbiota dysbiosis (e.g., excessive acidification of the gut) [145,146].

To support the concept of bidirectional communication between GM and muscle and vice versa, it should be noted that not only PE elicits an increase in gut SCFAs [134] but also that SCFA supplementation has been shown to improve the atrophy of muscle mass in GF mice [147], or reduce muscle atrophy in aging mice through butyrate treatment [148]. Both excessive PE and inactivity are not only detrimental to the integrity of muscle mass (e.g., via an increase in inflammation and oxidative stress), but also the alteration in GM diversity and GI dyshomeostasis induced by excessive exercise may contribute to systemic inflammation (e.g., via the production of bacterial toxins) and negatively impact protein synthesis and muscle mass [149]. The mutual interchange between GM and SM (the “gut–muscle” axis) is particularly well illustrated in the case of gut microbial ecosystem alteration. As a key player in the endocrine system, the insulin-like growth factor- (IGF-)1 is an anabolic factor critically involved both in the regulation/stimulation of muscle mass and in muscle atrophy [150]. From a mechanistic point of view, IGF-1 drives SM growth via the phosphatidylinositol 3-kinase (PI3K)/protein kinase B (Akt)/mTOR signaling pathway that, by increasing glucose transport, promotes protein synthesis and inhibits SM proteolysis [151]. The lack of GM, as in GF mice, causes a decrease in IGF-1 muscle expression and muscle atrophy, while either GM transplantation or SCFAs supplementation in GF mice appear able to induce a partial recovery of SM atrophy [147]. This is further corroborated by the positive effects observed after high-intensity exercise in athletes that underwent probiotic supplementation (i.e., *Lactobacillus plantarum*, PS128), which was also shown to increase the abundance of SCFAs as well as the abundance of the genera *Akkermansia*, *Bifidobacterium*, and *Lactobacillus* [152]. Interestingly, different factors such as non-strenuous PE and prebiotics/probiotics and SCFA supplementation [153] have in common the ability to improve SM metabolism (e.g., insulin sensitivity) and preserve SM mass during aging. In agreement, SCFA supplementation can improve exercise performance (i.e., endurance) in mice, thus demonstrating a direct effect of GM on muscle energy metabolism via an action mediated by SCFAs [153]. It is also known that inflammation is one of the major determinants underlying muscle catabolism and the loss of muscle mass [154], as demonstrated in chronic inflammatory states, such as obesity and insulin resistance [155]. Notably, restoring specific bacteria species such as *Lactobacillus* through probiotic supplementation (i.e., *L. reuteri* 100-23 and *L. gasseri* 311476) has been shown to reduce the muscle expression of atrophy markers and inflammatory cytokines, which were not affected by supplementation with other bacterial species [156].

Illustrative of the GM-to-muscle functional communication is the degree of GM diversity between young adults and older people (often suffering from dysbiosis) and the different liability to develop sarcopenia, which is an age-associated clinical condition characterized by the progressive loss of muscle mass and strength, as well as by a reduction in muscle fibers and a consequent decrease in physical performance [157]. Sarcopenic individuals show a reduced abundance of SCFA-producing bacteria such as *Faecalibacterium prausnitzii* [158]. The mechanistic influence of SCFAs on muscle metabolism is complex and may engage different direct and indirect routes. Butyrate can regulate insulin homeostasis (i.e., insulin sensitivity), promoting not only an increase in energy expenditure and muscle fatty oxidation [159] but also the inhibition of intramuscular lipid accumulation [148]. Also, SCFAs such as acetate and propionate have a favorable impact on both insulin-dependent and insulin-independent muscle glucose uptake [160], also eliciting (i.e., by acetate treatment) an increase in glucose transporter 4 (GLUT4) gene (*Glut4*) expression in SM [161]. In particular, acetate treatment has been shown to induce a higher expression of myosin heavy chain type I-isoform (MHCI) and MHC type-IIa isoform (MHCIIa), consequently increasing muscle oxidative capacity and endurance performance in mice [162]. However, while direct SCFAs administration can increase muscle fatty oxidation and endurance exercise, the dietary modulation of gut microbial composition may have opposite effects on muscle contractile capacity. In an ingenious study, one cohort of mice received a low microbiome-accessible carbohydrate (LMC) diet, while a second cohort was fed with a high microbiome-accessible carbohydrate (HMC) diet [163]. After a 6-week diet regimen, animals fed an LMC diet exhibited lower exercise capacity, reduced muscle mass (of the tibialis anterior), and decreased concentration of fecal and plasma SCFAs, while endurance capacity was restored by fecal microbiota transplantation (FMT) from HMC-fed donor mice and just one serving of fermentable fibers [163]. SCFAs can reach the ileum and colon and engage G protein-coupled receptors (GPCRs) 41 and 43, also known as the class of free fatty acids sensing receptors (FFARs), FFA2 (i.e., GPCR43) and FFA3 (i.e., GPCR41) [164,165]. Since both FFA2 and FFA3 are also expressed in adipose tissue (AT) and pancreatic β-cells, FFA2 and FFA3 also have a role in the regulation of insulin secretion and in the reduction in susceptibility to insulin resistance [166,167]. Importantly, the FFA2 receptor is involved in hormonal gut release, as evidenced by the control exerted on glucagon-like peptide 1 (GLP-1) secretion in response to SCFAs [168]. As further demonstrated, propionate is able to stimulate both peptide YY (PYY)- and GLP-1-signaling by FFA2 receptors expressed on colonic enteroendocrine L-cells, an effect repressed in FFA2 KO mice [169]. Apart from the modulation/suppression of appetite, the insulinotropic GLP-1 hormone is also involved in lowering glycemia (i.e., glucose metabolism) and in the regulation of insulin and glucagon secretion [170]. The GLP-1 receptor (GLP-1R) is expressed in many tissues, including the ENS, and in SM, where it participates in glycogen synthesis and in the potentiation of glucose uptake [171]. Remarkably, GLP-1 agonists and antidiabetic drugs such as exendin-4 have been shown to slow down SM atrophy via the recruitment of GLP-1R-mediated suppression of muscle atrophic factors [172]. A recent study also provided evidence that the in vivo overexpression of GLP-1 in SM is able to increase endurance capacity, as well as increase glycogen synthesis, glucose uptake, and the percentage of type I slow-twitch oxidative fibers [173]. Interestingly, this study corroborated the hypothesis that the GLP-1-dependent phosphorylation of AMP-activated protein kinase (AMPK) and increase in oxidative capacity represent possible mechanistic explanations underlying the improvement in endurance performance following GLP-1 overexpression [173]. However, AMPK phosphorylation can also be induced in SM by SCFAs such as acetic acid [174], thus activating AMPK targets such as peroxisome proliferator-activated receptor γ coactivator-1-α (PGC-1α), in turn promoting fatty acid oxidative metabolism [161,174]. An increase in exercise performance was also shown as a result of *Lactobacillus plantarum* (*L. plantarum*) supplementation, which increased muscle weight and exercise endurance and increased gastrocnemius type I fibers [175]. The development of GLP-1 resistance is observed in obese patients with type 2 diabetes, in which not only adiposity but also dysbiosis can play an instrumental role [176]. In other terms, GM is a major link between GLP-1 secretion, muscle function, and the beneficial effects of PA. Indeed, PE potentiates the effects of GLP-1 agonists via GM-derived SCFAs [177], and, as reminded, SCFAs can control and/or potentiate GLP-1 secretion [168] (Figure 3).

## 5. Skeletal Muscle-to-Gut Microbiota Axis

The reciprocal modulation between GM and SM is particularly corroborated when data can clearly demonstrate the effects of both PA and PE on the gut microbial ecosystem. As a whole, exercise is able to drive compositional changes to the gut microbial ecosystem, although significant differences may depend on the exercise intensity and duration, frequency, diet, and metabolic status of the subjects [178]. GM diversity changes with aging, but regular PE has been shown to induce the remodeling of gut microbial composition and abundance also in overweight individuals, as indicated by a large screening study performed on data including about 900 elderly subjects [179]. Notably, exercise training per se improves cardiometabolic/cardiorespiratory fitness in terms of higher peak oxygen (O_2_) capacity (VO_2_ max/kg), and such an increase in exercise/endurance capacity is responsible for more than 20% of the overall increase in microbial richness induced by PE, as well as the increase in butyrate production [136]. Well-trained running athletes were shown to develop a GM enrichment of the *Veillonella* genus after their marathon performance [180]. In the same study, the subsequent transplantation of isolated *Veillonella* in mice was shown to increase their endurance performance [180], an effect ascribed to the ability of the *Veillonella* genus to use lactate, which could be further metabolized into propionate and acetate, thus affecting endurance running. Exercise training, not only in elite athletes but also in sedentary subjects that start a period of PE, induces drastic changes in GM composition, functionality, the diversity of gut microorganisms, and increases the production of SCFAs [181,182]. Exercise training does not only modify the richness of GM composition but directly affects the production of SCFAs via increasing the abundance of SCFA-producing bacteria such as *Ruminococcaceae* and *Prevotella* [183]. The intestinal intra-epithelial lymphocytes (IELs) are considered as the sentinels of intestinal mucosal immunity [184], and some studies have demonstrated that PE can increase, at an IEL level, the secretion of anti-inflammatory cytokines such as IL-10 [185].

It is now well acknowledged that SM is an endocrine organ secreting multiple bioactive peptides following endogenous and exogenous stimuli, such as muscle contraction and diet-derived nutrients [186]. For instance, it is now recognized that during PE and muscle contraction, there is a robust increase in IL-6 in the bloodstream [187]. IL-6 is one of the first identified myokines, and, as far as we know, an increase in IL-6 in the bloodstream is associated with obesity and type 2 diabetes (i.e., insulin resistance) [188]. Additionally, it is associated with AMPK-dependent fat oxidation and glucose uptake after muscle contraction [189]. Notably, the increase in IL-6 after muscle contraction produces a direct stimulatory effect on GLP-1 secretion from enteroendocrine L-cells, improving glucose metabolism and insulin sensitivity [190]. Both SCFAs and PE can induce the transcription and expression of PGC-1α in muscle. PGC-1α expression is triggered by muscle contraction and is required for muscle fiber type switching towards the oxidative phenotype, thus adapting muscle capacity and metabolism to endurance exercise [161,174,191].

### 5.1. Irisin and BDNF Are Paradigmatic Myokines Linking Physical Activity to the Shaping of Gut Microbiota

Irisin is a recently described PGC-1 alpha dependent molecule. Indeed, PGC-1α is required for the gene expression and synthesis of a transmembrane protein called fibronectin type III domain-containing protein 5 (FNDC5), the cleavage of which produces a peptide named irisin [192]. Acting in a hormone-like fashion, the role of irisin is particularly intriguing in this context. Indeed, irisin secretion is elicited by muscle contraction, and, for this reason, irisin is considered an exercise-induced myokine; its level in the bloodstream depends almost entirely on muscle tissue [192]. However, beyond its stimulatory effects on energy metabolism (thermogenesis), adipose tissue (browning white fat depots), and bone (increase in bone mineral density), irisin can be regarded as a bridge between the muscle and brain. Irisin can be found in the brain (both neurons and glial cells) and in different brain areas including the hippocampus, cortex, hypothalamus, and cerebellum [193,194,195]. Surprisingly, endurance exercise induces the neural expression of the PGC-1α/FNDC5/irisin pathway, which, in turn, can promote adult hippocampal neurogenesis (AHN) via an increase in brain-derived neurotrophic factor (BDNF) [194]. BDNF signaling is involved in synaptic plasticity (e.g., long-term potentiation, LTP), and this neurotrophin is required for cognitive function, especially within the hippocampus [196]. Notably, BDNF is also secreted upon training exercise and for this reason is also considered a brain-targeting myokine, as illustrated by exercise-induced BDNF expression in the hippocampus and improvements in neuroplasticity and spatial memory [197,198]. Several studies have also uncovered an important irisin-mediated neuroprotective potential by showing, for instance, that irisin treatment protects the BBB [199] or hippocampal neurons from apoptosis [200] under different ischemic injuries in rats (e.g., middle cerebral artery occlusion), and that the increased expression of BDNF can mediate these neuroprotective effects [201].

For the time being, the evidence that irisin can provide many of its multiple beneficial effects due to a mechanism(s) involving changes in the GM is only indirect, but nevertheless noteworthy. Until recently, the protective anti-inflammatory potential of irisin was never investigated in morbid medical conditions in which gut dysbiosis plays a pathogenetic role, such as ulcerative colitis (UC). By studying the effects of irisin administration in a mouse model of UC, an improvement in the inflammatory status involving decreases in IL-12 and IL-23 plasma levels [202], both interleukins involved in the amplification and differentiation of the T-helper (Th) type 1 (Th1) and type 17 (Th17) lymphocyte response [203,204], and the reversal of gut microbial communities that were found abnormally represented in UC mice (e.g., *Bacteroides* and Lactobacillaceae) were shown. Another recent study has demonstrated that irisin administration can improve the integrity of the gut mucosal barrier and can counterbalance the gut dysbiosis induced in rats that underwent an experimental model of myocardial ischemia–reperfusion injury [205]. Equally illustrative of the capacity demonstrated by irisin to attenuate intestinal damage is one study in which irisin administration was used to reduce injury to epithelial cells, as well as apoptosis and oxidative stress in a mouse model of acute pancreatitis [206]. Interestingly, not only irisin but also BDNF has been shown to have a prominent role in the modulation of GM biodiversity and in the homeostasis of the intestinal mucosal barrier. Indeed, BDNF KO mice (BDNF^−/−^) and wild-type mice (BDNF^+/+^) were shown to basically differ from each other in the ultrastructure of their colonic epithelium, which was impaired in mice lacking BDNF [207]. BDNF^−/−^ mice displayed a selective decrease in epithelial tight junction proteins, such as zonula occludens-1 and occludin, which expand the vulnerability of the intestinal mucosal barrier [207].

### 5.2. Irisin and BDNF as “Ideal” Players Connecting Physical Activity and Risk of Developing AD

Both endurance (i.e., aerobic) and resistance (i.e., strength) exercise up-regulate FNDC5 gene expression and irisin circulating levels in mice and humans [208,209], and training exercise has been systematically used to investigate the relationship between PA, FNDC5/irisin, and BDNF expression within the hippocampus [194]. Remarkably, FNDC5 neural expression is reduced in the hippocampus of AD-like mice, in which the adenovirus-associated brain delivery of FNDC5/irisin has been shown to offset memory deficits and improve neural plasticity (e.g., LTP) [210]. Thus, this study elegantly suggests that the positive effects provided in AD patients by PE can be mediated by irisin signaling. Not less important, in a study on cultured astrocytes, it was shown that irisin exerts a neuroprotective effect against β-amyloid-induced toxicity and cell injury, and that the neuroprotection provided by irisin regarding neuron viability could be attributed to the blockage of the astrocyte release of pro-inflammatory cytokines such as IL-1β and COX-2 via a decrease in the NF-κB pathway [211]. In agreement with this, age-associated cognitive decline has been shown to be counteracted by exercise-induced activation of the PGC-1α/FNDC5/irisin signaling pathway [212]. BDNF levels in the cerebrospinal fluid (CSF) have been shown to predict the progression from the MCI status to AD diagnosis [213], and PA-elicited irisin-BDNF activation is essential for AHN [194]. Remarkably, a study published in 2018 [214] provided a striking demonstration that the stimulation of AHN is necessary but not sufficient to rescue the cognitive deficits exhibited by the AD-like 5xFAD mouse model, while inducing AHN. In parallel, simulating training exercise through a pharmacological increase in brain BDNF levels can produce an improvement in cognitive function (i.e., working and spatial memory) in AD-like mice. It should also be noted that adult neurogenesis can be tightly regulated by changes in GM composition. For instance, a diet-induced alteration in GM and a reduction in SCFA production can lead to a decrease in neural stem cells and neural progenitor cells (which are the “engine” of adult neurogenesis in specialized niches such as the ventricular-subventricular zone (V-SVZ) and the subgranular zone (SGZ) of the hippocampal dentate gyrus), while SCFA-treated mice showed increased brain plasticity and neurogenesis [215,216], including BDNF expression [217]. Hence, if GM can regulate neurogenesis and BDNF levels along the GM–brain axis through the parasympathetic system and the vagus nerve pathway [218], we also know that exercise training does not only change GM diversity [179,181,183] but also directly increases BDNF expression in the hippocampus [197,198,219]. The crosstalk between BDNF and irisin in the brain appears to be tightly regulated. Indeed, BDNF expression appears down-regulated in the brains of APP/PS1 transgenic AD-like mice [220]. In the same study, Aβ_1-42_ oligomers decreased FNDC5 expression in neuro-2a (n2a) mouse neuroblasts and induced a suppressive effect on BDNF expression that was reversed by FNDC5 overexpression, and intranasal BDNF delivery was shown to reduce Aβ aggregation and cognitive decline in the brains of APP/PS1 AD-like mice [220].

From this view, the growing body of evidence supporting the beneficial effects of PE against the risk of developing AD can be explained at multiple levels of mechanistic interaction by which training exercise appears to affect the downstream gut microbial ecosystem and upstream neural plasticity, anti-inflammatory signaling pathways, neurotrophin expression, and neurogenesis via the production of myokines such as irisin and BDNF. Exercise training can modify selected bacterial strains, as evidenced in overweight women assigned to aerobic exercise in which weight loss was associated with an increase in *Bifidobacterium* and *Lactobacillus* populations [221]. Notably, probiotic supplementation with *Lactobacillus plantarum* and *Bifidobacterium bifidum* in AD-like mice that underwent concomitant exercise training was demonstrated as an effective strategy to improve memory performance and increase hippocampal BDNF expression [222].

### 5.3. Irisin, BDNF, and the Anxiety–Depression Spectrum: Muscle–Gut–Brain Axis and Non-Cognitive Symptoms in Alzheimer’s Disease

As outlined in the introductory section, NPSs such as anxiety (e.g., excessive worry, irritability) and depression (e.g., social withdrawal) are frequently observed and diagnosed during AD progression. Although a strong pathogenetic explanation for the incidence of NPSs in AD is yet to be provided, the functional connection between the muscle, gut, and brain we delineated (i.e., muscle–gut–brain axis) can suggest some novel points of discussion and future matters for investigation. It is worth considering that both irisin and BDNF have been shown to possess an anti-depressive potential, and this has received substantial and repeated confirmation. In brief, irisin and BDNF plasma levels are reduced in depressed subjects [223,224], and the direct brain infusion of irisin or BDNF has been shown to suppress depression-like behaviors in a similar fashion [225]. The impact of resistance exercise training in depressed subjects has been addressed by systematic reviews and large meta-analysis studies that have further validated the idea of a significant association between PE and a decrease in depressive symptoms, as well as the idea of exercise training as an adjuvant therapy for depression [226,227]. Moreover, while the role of alterations in GM composition in depression pathogenesis is currently recognized [228], and fecal microbiome transplantation is documented as a feasible non-pharmacological adjunctive therapy for depression [229], it is interesting to ascertain whether exercise can reshape the GM community and produce positive effects in depressed subjects. A recent randomized controlled trial has reported an alteration in the profile of the gut microbial population with a parallel improvement in depressive symptoms in young adolescents who underwent a program involving an exercise training intervention [230]. Some of the bacterial genera that were increased in depressed adolescents following a program of exercise intervention (e.g., *Coprococcus* and *Blautia* from *Firmicutes* phylum) [230] have also been described to be depleted in depression [231], although a decrease in butyrate-producing genera such as Coprococcus has been found in patients with Lewy body dementia [232] and in AD-like mice [233]. Moreover, an increase in *Blautia* has been found to be causally associated with a reduced risk of AD [234], and a 12-week mixed exercise training was reported to produce an increase in some genera such as *Blautia*, *Dialister*, and *Roseburia* [235].

If depression is associated with metabolic diseases [236], and regular exercise is an anti-depressant factor for patients with major depression, and irisin is an exercise-dependent myokine showing an anti-depressant activity [224,225], then irisin can achieve its anti-depressant effects via the regulation of mechanisms involved in energy metabolism. The fact that irisin could improve depression-like symptoms by impinging on energy metabolism, and, in particular, by increasing levels of enzymes involved in glucose metabolism, transport (i.e., type I and type II hexokinase), and uptake in astrocyte cells has been previously reported [237]. Defective glucose metabolism and insulin resistance (IR) are associated with depression [238], and recent reports have evidenced that the risk of major depressive disorder can be reliably predicted either by longitudinal time-spaced measures of IR or by using impaired glucose metabolism as biomarker to implement a machine learning-based model of prediction [239]. Exercise-induced irisin can impact the function of GLUT4 and improve glucose uptake and IR by stimulating GLUT4 translocation towards the membrane of SM cells (i.e., myocytes) [240]. Considering also that both high-intensity interval training and moderate-intensity continuous training can increase irisin secretion and GLUT4 mRNA expression [241], one of the mechanisms by which irisin can achieve its anti-depressant action is through an improvement in glucose uptake. In a recent scoping review [242] focused on key bacteria involved in glucose metabolism, the authors identified 45 bacterial taxa for which there is evidence of an inverse relationship with hyperglycemia (fasting glucose) and IR. In particular, *A. muciniphila*, *Bifidobacterium longum*, *Clostridium leptum* group, *Faecalibacterium prausnitzii*, and *Faecalibacterium* were the taxa selected as relevant for the study of prevention strategies against metabolic diseases [242]. For instance, *Faecalibacterium prausnitzii* is a butyrate-producing bacteria inversely associated with type II diabetes, also described as a therapeutic option to decrease fasting glucose and IR in diabetic patients [243]. Thus, SCFA-producing bacteria (such as *A. muciniphila*) might be a mechanistic link between exercise-inducing irisin and BDNF and the positive impact on NPSs in AD. From this view, future investigations should redirect their aims at determining additional mechanisms by which irisin contributes to glucose homeostasis by decreasing hyperlipidemia, fasting glucose, and IR via its action on SCFA-producing bacteria to improve depressive symptoms in AD patients (Figure 4).

## 6. Conclusions and Limitations

NPSs, such as depression and apathy, commonly occur in AD patients and are associated with accelerated progression of the pathology, as well as poorer quality of life and increased caregiver burden. Interestingly, regular PE and aerobic training have been found to have neuroprotective benefits in delaying the onset and progression of AD, likely through enhancing vascularization and neurogenesis, increasing brain volume, elevating neurotrophic factors, reducing inflammation and oxidative stress, and improving cognitive functions such as executive functions. In recent years, gut dysbiosis has been investigated in AD patients since it may contribute to neuroinflammation and cognitive decline through diverse mechanisms, including increased gut permeability, higher production of LPS and pro-inflammatory cytokines, and the dysregulation of neurotransmitter function and tryptophan metabolism. Studies in animal models of AD also support a role for GM in modulating amyloid pathology and cognitive function. Importantly, the GM and SM engage in a bidirectional communication via SCFAs and metabolites like lactate, with microbe-derived SCFAs playing an important role in regulating muscle mass, metabolism, and performance through multiple mechanisms, including modulating insulin sensitivity, glycogen synthesis, glucose uptake, fiber composition, and mitochondrial function. Changes in GM composition through diet or probiotics can therefore impact muscle health and PE capacity.

PE can reshape the GM composition as well through increasing microbial diversity, the abundance of butyrate-producing bacteria, and SCFA production, which in turn contribute to improving cardiorespiratory fitness and endurance performance. Exercise also induces the secretion of myokines like IL-6 from SM that can stimulate GLP-1 secretion from the gut, while both SCFAs and PE up-regulate PGC-1α expression in muscle to promote an oxidative phenotype that supports endurance exercise. The myokine irisin may exert neuroprotective effects possibly by increasing BDNF expression, and recent evidence indicates that irisin may help to restore GM dysbiosis and improve intestinal barrier integrity in various disease models. Additionally, BDNF signaling appears important for maintaining colonic epithelial integrity, suggesting that exercise-induced myokines like irisin and BDNF may contribute to modulating gut–brain interactions.

In conclusion, exercise could reduce AD risk even by upregulating the production of myokines such as irisin and BDNF that can stimulate hippocampal neurogenesis, modulate inflammation, and reshape the GM to increase beneficial bacteria and SCFA production—effects that potentially synergize to improve cognition, neural plasticity, and gut–brain interactions in AD. Finally, exercise-induced irisin and BDNF could help to alleviate NPSs in AD by improving glucose metabolism and IR, and by modulating gut bacteria involved in glucose regulation, especially SCFA-producing bacteria which are often reduced in depression. This knowledge can be of great value in different clinical settings in which very innovative non-pharmacological strategies of intervention can be implemented in real-life situations. For instance, eligible AD patients could be recruited in specific exercise-focused rehabilitation programs with the parallel quantification of irisin blood levels and GM analysis longitudinally at different time points. Meanwhile, the same AD patients could be screened for the expression of NPSs and the use/prescription of antipsychotic medications, with a focus on gender-specific differences that, compared with other fields of medicine [244,245,246], are currently often neglected.

However, given the large availability of mouse data over studies performed on human subjects, it should be observed that conclusions, perspectives, and interpretations are inevitably based on results obtained using animal models and laboratory rodents. Thus, caution should be taken in making general assumptions, as well as in translating findings from animal models to humans. For this reason, we hope that additional investigation on human subjects in the next future could help us to gain further insights into the mechanisms involved in the functional crosstalk between gut–muscle–brain and muscle–gut–brain that we discussed here.

Considering the potential limitations in the use of PE to improve brain health, it is essential to recognize that although exercise has shown promise for enhancing cognitive function, its efficacy can vary widely among individuals. Factors such as age, baseline cognitive status, and the presence of neurodegenerative diseases can influence the extent of cognitive benefits derived from PE. Additionally, the optimal type, intensity, and duration of exercise required to achieve specific cognitive outcomes remain an area of ongoing research and debate. Moreover, adherence to exercise regimens over the long-term can be challenging for many individuals, particularly older adults, and those with cognitive impairments, which can limit the potential benefits. Indeed, one important limitation lies in the difficulty of implementing rigorous exercise protocols, particularly among elderly individuals and AD patients, due to mobility deficits and cognitive impairments limiting adherence to well-organized exercise regimens. Thus, a comprehensive approach that considers not only the biological mechanisms but also the practical aspects of exercise interventions, especially in vulnerable populations, is particularly required to yield a more exhaustive understanding of the intricate interplay between PE, neurodegenerative diseases, aging, and the GM. New knowledge on this subject can inspire the development of tailored interventions, taking advantage of the tight molecular and functional interplay between PE and GM, with the aim to delay cognitive decline and positively impact NPSs in AD patients. Global efforts to prevent AD and dementia underscore the need for accessible, cost-effective strategies adaptable to different socio-economic and cultural environments. While modifiable risk factors have been identified, research has predominantly focused on high-income countries, despite the growing dementia crisis in low- and middle-income nations. A multidomain preventive approach, inspired by successful models for other chronic conditions, holds promise for addressing the intricate nature of cognitive impairment and offers scalability, supported by eHealth tools and personalized interventions [247]. International collaboration is the key to identifying preventive strategies, potentially including pharmacological interventions, to address the global dementia challenge.

From this view, the combination of PE with the use of wearable technologies, and the concomitant assessment of PE-derived messengers (e.g., irisin and BDNF), together with the monitoring of GM dysbiosis and probiotic/SCFA supplementation can represent a disease-modifying strategy for adjuvant therapy within the future perspective, with hopes to slow down cognitive decline and reduce the use of antipsychotic medications.

## Figures and Tables

**Figure 1 ijms-24-14686-f001:**
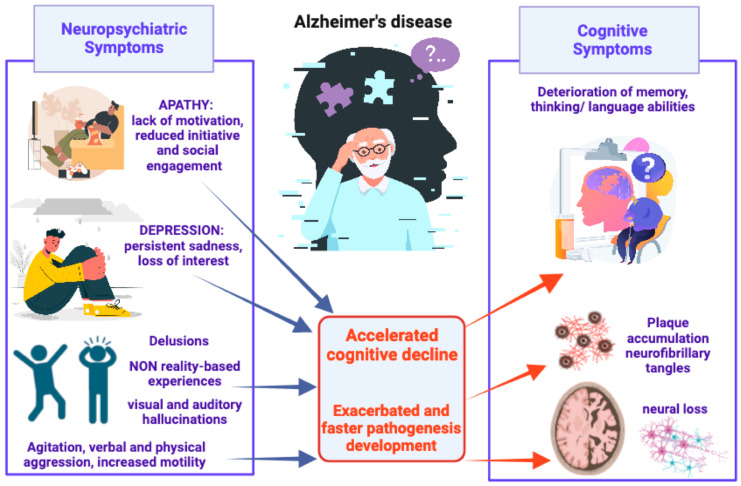
Schematic representation of AD-associated NPSs, from most to least common, and their detrimental impact on pathogenesis development and cognitive decline.

**Figure 2 ijms-24-14686-f002:**
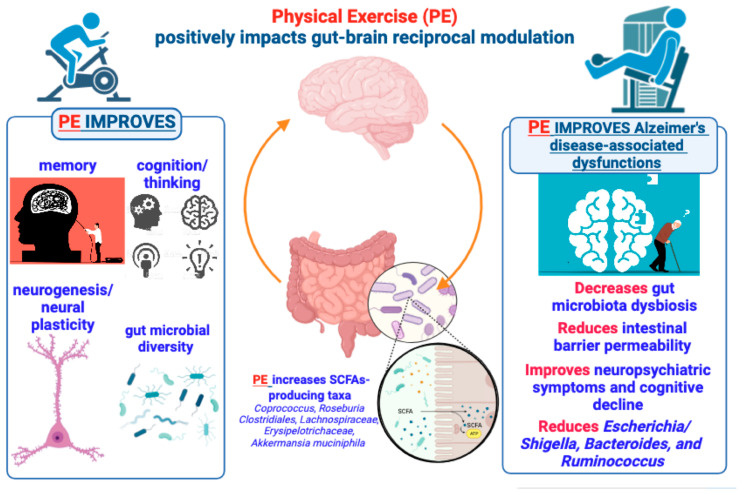
Brief summary of the positive effects produced by PE on brain functions and gut microbiome underlying the gut–brain bidirectional crosstalk. According to the text, the figure illustrates and suggests that PE can produce beneficial effects on brain functions via a gut microbiota (GM)-mediated action, so that a PE–GM–brain axis may have a crucial role in preventing or delaying AD development and symptomatology.

**Figure 3 ijms-24-14686-f003:**
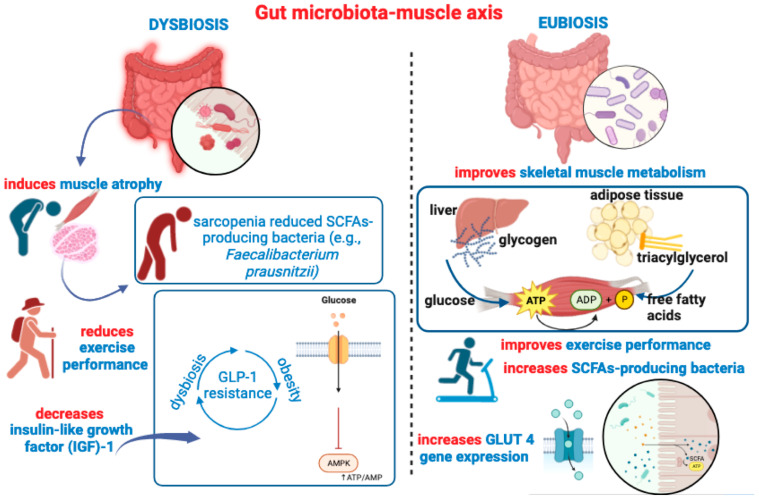
This figure recapitulates main ideas outlined in Section 4, depicting the detrimental impact of gut dysbiosis on muscle health, inducing atrophy, decreasing IGF-1 expression, and reducing gut colonization of SCFA-producing bacteria. A pathogenetic vicious circle may be triggered between gut dysbiosis and GLP-1 resistance, contributing to altered glucose metabolism, decreased AMPK phosphorylation, and deficient motor-exercise performance. On the right side of the figure, is depicted the beneficial impact of a healthy “eubiotic” GM, which can also be achieved through SCFAs or probiotic supplementation. Many beneficial effects of SCFAs on muscle health, such as oxidative capacity, are also illustrated.

**Figure 4 ijms-24-14686-f004:**
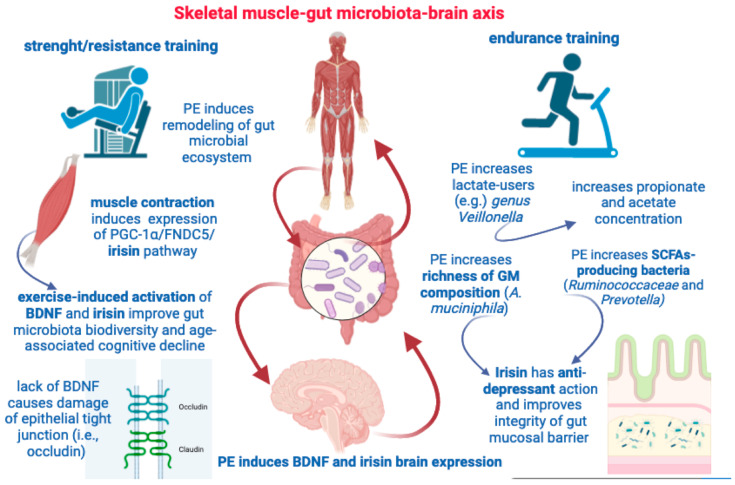
This figure illustrates some key aspects highlighted by the muscle–gut–brain axis. Both resistance and endurance training are able to induce remodeling of gut microbial population and increase SCFAs concentration. Muscle contraction, per se, can induce irisin and BDNF expression with beneficial effects on GM diversity and cognitive deterioration. PE can increase GM richness, brain expression of BDNF and irisin, and provide both anti-depressant effects and protect the gut mucosal barrier.

## Data Availability

Not applicable.

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
