# Peer review of "Physical Exercise as Disease-Modifying Alternative against Alzheimer’s Disease: A Gut–Muscle–Brain Partnership"

_ijms, 2023, doi:10.3390/ijms241914686_

Round 1
Reviewer 1 Report
The article addresses important aspects of both Physical Exercise and Physical Activity that elicit beneficial effects for delaying the progression of AD. The approach considering the gut-muscle-brain axis is extremely important to guide new and larger investigations on the positive effects of EF/AF in the treatment of human diseases.
Below are my considerations:
This reviewer suggests to replace “option” for “alternative” in the title.
Throughout the text, I noticed that some acronyms are conceptualized repeatedly as in lines 73 and 178.
Lines 82-83. This sentence needs a reference.
Lines 110 to 112, you mention that many studies have found a correlation between agitation and loss of brain volume in various brain regions, but you only cite a single article (citation 31), and that doesn't seem to say the same as your sentence. please review this.
Line 181 to 184, this reviewer suggests including more recent literature such as: a) Kaur, D., Sharma, V. & Deshmukh, R. Activation of microglia and astrocytes: a roadway to neuroinflammation and Alzheimer’s disease. Inflammopharmacol 27, 663–677 (2019). https://doi.org/10.1007/s10787-019-00580-x; e b) Rajesh, Y.; Kanneganti, T.-D. Innate Immune Cell Death in Neuroinflammation and Alzheimer’s Disease. Cells 2022, 11, 1885. https://doi.org/10.3390/cells11121885
Line 183, is the term “overactivation” appropriate?
Line 246, the authors mention that different studies reported an increase in the presence of certain bacterial genera, but at the end of the sentence only one study is cited.
The sentence contained in lines 298 and 299 seems to be confused. Please clarify what aspects you refer to the implication of the BBB.
In topic 3.4, it seems that the terms “physical exercise, PE” and “physical activity, PA” are confused (lines 338 to 340). In other parts of the text, the terms seem to be treated as the same thing (which is not true) or are addressed together in the same paragraph or sentence without considering the particularities and/or limits between the two terms.
The sentences in lines 360 to 364 should be redone. In line 360, what does “inactivity” refer to?
By logical order, the conclusions chapter should be topic “6”.
The text has a few grammatical mistakes. Please consider reviewing them.
In conclusion, the authors should include a few paragraphs about limitations and caveats considering the use of exercise aimed at brain health.
About the figures: Figures 1, 2 and 3 lack key connections that integrate the elements so that they make logical sense.
this reviewer suggests a review by native speaker
Author Response
This reviewer suggests to replace “option” for “alternative” in the title.
Answer:We thank the reviewer for this recommendation. By this replacement, the title looks now more attractive and sharper.
Throughout the text, I noticed that some acronyms are conceptualized repeatedly as in lines 73 and 178. Answer: Right. We have now adjusted the use of acronyms.
Lines 82-83. This sentence needs a reference. Answer: Following this advice, we have now inserted a new reference in line with that sentence. Majer, R., Adeyi, O., Bagoly, Z., Simon, V., Csiba, L., Kardos, L., Hortobágyi, T. & Frecska, E. (2020). Neuropsychiatric symptoms, quality of life and caregivers’ burden in dementia. Open Medicine, 15(1), 905-914. https://doi.org/10.1515/med-2020-0124.
Lines 110 to 112, you mention that many studies have found a correlation between agitation and loss of brain volume in various brain regions, but you only cite a single article (citation 31), and that doesn't seem to say the same as your sentence. please review this. Answer: Right. Hence, we have now reviewed this issue removing the previous reference and adding three new references, as follows: 1) Bateman, D. R., et al., Neuropsychiatric Syndromes Professional Interest Area (NPS‐PIA) (2020). Agitation and impulsivity in mid and late life as possible risk markers for incident dementia. Alzheimer's & dementia (New York, N. Y.), 6(1), e12016. https://doi.org/10.1002/trc2.12016; 2) Rosenberg, P. B., et al., 2015. Neuropsychiatric symptoms in Alzheimer's disease: What might be associated brain circuits?. Molecular aspects of medicine, 43-44, 25–37. https://doi.org/10.1016/j.mam.2015.05.005; 3) Nowrangi, M., et al., 2021. The association of neuropsychiatric symptoms with regional brain volumes from patients in a tertiary multi-disciplinary memory clinic. International Psychogeriatrics, 33(3), 233-244. doi:10.1017/S1041610220000113.
Line 181 to 184, this reviewer suggests including more recent literature such as: a) Kaur, D., Sharma, V. & Deshmukh, R. Activation of microglia and astrocytes: a roadway to neuroinflammation and Alzheimer’s disease. Inflammopharmacol 27, 663–677 (2019). https://doi.org/10.1007/s10787-019-00580-x; e b) Rajesh, Y.; Kanneganti, T.-D. Innate Immune Cell Death in Neuroinflammation and Alzheimer’s Disease. Cells 2022, 11, 1885. https://doi.org/10.3390/cells11121885 Answer: we added the recommended references.
Line 183, is the term “overactivation” appropriate? Answer: we do believe that the term “overactivation” is appropriate in this context; as a matter of fact, other authors have used it. See, among others: 1) Galvani, G., et al., 2021) Inhibition of microglia overactivation restores neuronal survival in a mouse model of CDKL5 deficiency disorder. Journal of neuroinflammation, 18(1), 155. https://doi.org/10.1186/s12974-021-02204-0; 2) Pomilio, C., et al., 2020. Microglial autophagy is impaired by prolonged exposure to β-amyloid peptides: evidence from experimental models and Alzheimer’s disease patients. Geroscience, 42, 613-632; 3) Wang, J., et al., 2018. Supplementation of lycopene attenuates lipopolysaccharide-induced amyloidogenesis and cognitive impairments via mediating neuroinflammation and oxidative stress. The Journal of nutritional biochemistry, 56, 16-25.
Line 246, the authors mention that different studies reported an increase in the presence of certain bacterial genera, but at the end of the sentence only one study is cited. Answer: we added two extra references to further support the statement. 1) Chen, Y., Xu, J., & Chen, Y. (2021). Regulation of Neurotransmitters by the Gut Microbiota and Effects on Cognition in Neurological Disorders. Nutrients, 13(6), 2099. https://doi.org/10.3390/nu13062099; 2) Molinero, N., et al., 2023. Gut Mi-crobiota, an Additional Hallmark of Human Aging and Neurodegeneration. Neuroscience, 518, 141–161; https://doi.org/10.1016/j.neuroscience.2023.02.014.
The sentence contained in lines 298 and 299 seems to be confused. Please clarify what aspects you refer to the implication of the BBB. Answer: To better clarify this aspect, we modified the sentence as follows: “Although the authors suggest a possible anti-inflammatory mechanism exerted by probiotics in alleviating depressive symptoms, there are still some unresolved problems, such as the implication of BBB damage or intestinal hyperpermeability for the evaluation of this effect.” (lines 299–302).
In topic 3.4, it seems that the terms “physical exercise, PE” and “physical activity, PA” are confused (lines 338 to 340). In other parts of the text, the terms seem to be treated as the same thing (which is not true) or are addressed together in the same paragraph or sentence without considering the particularities and/or limits between the two terms. Answer: we agree with the reviewer. Indeed, the distinction between PE and PA is reported at the beginning of the manuscript. Thus, we revised the text thoroughly for possible inconsistencies. Many amendments have been made, and PA replaced with PE (or vice versa) according to the use made by the authors of the study cited.
The sentences in lines 360 to 364 should be redone. In line 360, what does “inactivity” refer to? Answer: The term “inactivity” has been replaced with “sedentary lifestyle”. Moreover, some lines have been rephrased.
By logical order, the conclusions chapter should be topic “6”. Answer: Right. We have now renumbered the “Conclusions and Limitations” section as paragraph 6.
The text has a few grammatical mistakes. Please consider reviewing them. Answer: The manuscript has been now proofread by a native speaker, and the grammatical mistakes found in the document amended.
In conclusion, the authors should include a few paragraphs about limitations and caveats considering the use of exercise aimed at brain health. Answer: as suggested, we described several limitations of the studies examined, and decided to include them in the sixth paragraph "Conclusions and limitations".
About the figures: Figures 1, 2 and 3 lack key connections that integrate the elements so that they make logical sense. Answer: by carefully revising these figures, we have paid attention in reshape them in several aspects including graphical features and clearer logical connections between the different elements portrayed, as suggested.
Reviewer 2 Report
Review
This work discusses the potential relationship between regular physical exercises (PE), gut microbiota (GM), and their impact on Alzheimer's disease (AD) and related neuropsychiatric symptoms (NPSs). It suggests that exercise may have neuroprotective benefits, delaying the onset and progression of AD, and improving cognitive functions. The authors also explore the role of gut dysbiosis in AD and its potential contribution to neuroinflammation and cognitive decline.
The work highlights the bidirectional communication between GM and skeletal muscle via short-chain fatty acids (SCFAs) and metabolites like lactate. The microbial composition can be influenced by exercise, leading to increased microbial diversity and SCFAs production, which may improve cardiorespiratory fitness and endurance.
The paper proposes that exercise-induced myokines, such as irisin and brain-derived neurotrophic factor (BDNF), may play a role in neuroprotection, influencing hippocampal neurogenesis, modulating inflammation, and potentially impacting gut-brain interactions. Additionally, myokines like irisin and BDNF could help alleviate NPSs in AD by improving glucose metabolism and insulin resistance through the modulation of gut bacteria involved in glucose regulation.
The conclusion suggests that exercise, myokines, and gut microbiota might synergistically improve cognition, neural plasticity, and gut-brain interactions in AD. The authors propose that further research is needed to delve deeper into the mechanisms of the gut-muscle-brain and muscle-gut-brain partnership outlined in the paper. Understanding these mechanisms could inspire tailored interventions to leverage the interplay between PE and GM to improve cognitive decline and NPSs in AD patients.
I really appreciate this paper for several reasons. Firstly, it explores a universal approach to curating neurological disorders, which offers significant advantages over pharmacological treatments. This approach enhances general health, providing benefits for alleviating other concurrent diseases, and it lacks deleterious side-effects. Secondly, despite focusing on general health, the authors delve into the molecular level with meticulous detail to explain the beneficial effects.
I believe that this review can be published in IJMS. However, there are several recommendations, which can improve the paper.
1. The weakness of the paper lies in its heavy reliance on mouse data. While all interspecies (in this case, interorder) differences are mostly unknown, they can be significant, particularly when considering the brain, which evolved very rapidly in primate lineage. Another notable difference is body size.
A recent study found that genes with the most significantly altered expression in evolution exhibited consistent unidirected changes in different organs, including the brain (referred to as 'unidirectionally changed genes', UCG). Significantly, even in the comparison between humans and primates, the UCG were enriched in the genes involved in neurological disorders (https://doi.org/10.1016/j.biosystems.2020.104256).
The usage of mouse data is unavoidable due to the abundance of available data for this model, which are lacking in humans. However, readers should be cautioned. In my view, the authors should acknowledge the above-mentioned paper and caution readers about the potential challenges in translating findings from animal models to humans.
2. It would be good to discuss the detrimental effects of high lactate content in the gut, which deteriorates muscle and heart function because of microbiota dysbiosis. Recent research has revealed that lactose intolerance caused by lactase overload leads to significant and lasting effects such as muscle and heart atrophy, and extensive transcriptome-wide rearrangements in rats (https://doi.org/10.3390/ijms24087063). These detrimental effects can be attributed to the excessive acidification of the gut (and consequently the whole body) originating from microbiota dysbiosis, which cause an overproduction of lactate.
3. A minor point to improve the paper would be to include a list of abbreviations after the list of keywords.
Congratulations to Dera Authors!
This Review will be in golgen collecion of IJMS. It will attract a lot of citations.
With admiration, Reviewer.
Author Response
We would like to thank the reviewer for his/her appreciation and the positive considerations expressed about our work and the ideas underlying it. Notably, the reviewer caught one important aspect of our paper, in which mice-based data are indeed overrepresented. On one hand, as noted, given the profusion of data focused on mice models this type of knowledge becomes somehow unescapable but, on the other hand, we considered the reviewer’s suggestion and inserted a novel (final) paragraph to warn the readers about the risk to draw conclusions mostly from results obtained on rodents.
2. It would be good to discuss the detrimental effects of high lactate content in the gut, which deteriorates muscle and heart function because of microbiota dysbiosis. Recent research has revealed that lactose intolerance caused by lactase overload leads to significant and lasting effects such as muscle and heart atrophy, and extensive transcriptome-wide rearrangements in rats (https://doi.org/10.3390/ijms24087063). These detrimental effects can be attributed to the excessive acidification of the gut (and consequently the whole body) originating from microbiota dysbiosis, which cause an overproduction of lactate. Answer: As suggested, we briefly discussed the detrimental effects of excessive lactate content in the gut (lines 357–359).
- 3. A minor point to improve the paper would be to include a list of abbreviations after the list of keywords. Answer: We included la list of abbreviations prior to the References as found in other articles published in IJMS.